# Investigating Whether the Mediterranean Dietary Pattern Is Integrated in Routine Dietetic Practice for Management of Chronic Conditions: A National Survey of Dietitians

**DOI:** 10.3390/nu12113395

**Published:** 2020-11-04

**Authors:** Hannah L. Mayr, Sarah P. Kostjasyn, Katrina L. Campbell, Michelle Palmer, Ingrid J. Hickman

**Affiliations:** 1Nutrition and Dietetics Department, Princess Alexandra Hospital, Brisbane, QLD 4102, Australia; katrina.campbell@health.qld.gov.au (K.L.C.); i.hickman@uq.edu.au (I.J.H.); 2Bond University Nutrition and Dietetics Research Group, Faculty of Health Sciences and Medicine, Bond University, Gold Coast, QLD 4226, Australia; sarahkostjasyn@hotmail.com; 3School of Allied Health, Human Services and Sport, La Trobe University, Melbourne, VIC 3086, Australia; 4Faculty of Medicine, University of Queensland, Brisbane, QLD 4072, Australia; 5Nutrition and Dietetics Department, Logan Hospital, Meadowbrook, QLD 4131, Australia; michelle.palmer@health.qld.gov.au

**Keywords:** Mediterranean diet, chronic disease, online survey, dietitian, barriers and facilitators, health services research

## Abstract

Evidence supports recommending the Mediterranean dietary pattern (MDP) in the management of cardiovascular disease (CVD), type 2 diabetes (T2D), non-alcoholic fatty liver disease (NAFLD) and solid organ transplant (SOT). However, the evidence-practice gap is unclear within non-Mediterranean countries. We investigated integration of MDP in Australian dietetic practice, and barriers and enablers to MDP implementation for chronic disease management. Dietitians managing CVD, T2D, NAFLD and/or SOT patients (*n* = 182, 97% female) completed an online survey in November 2019. Fewer than 50% of participants counsel patients with CVD (48%), T2D (26%), NAFLD (31%) and SOT (0–33%) on MDP in majority of their practice. MDP principles always recommended by >50% of participants were promoting vegetables and fruit and limiting processed foods and sugary drinks. Principles recommended sometimes, rarely or never by >50% of participants included limiting red meat and including tomatoes, onion/garlic and liberal extra virgin olive oil. Barriers to counselling on MDP included consultation time and competing priorities. Access to evidence, professional development and education resources were identified enablers. An evidence-practice gap in Australian dietetic practice exists with <50% of participants routinely counselling relevant patient groups on MDP. Strategies to support dietitians to counsel complex patients on MDP within limited consultations are needed.

## 1. Introduction

Chronic diseases are the leading cause of global burden of disease [1]. Highly prevalent chronic diseases include cardiovascular disease (CVD), type 2 diabetes (T2D) and non-alcoholic fatty liver disease (NAFLD), which have interrelated cardiometabolic risk factors and morbidity [2,3]. Poor diet quality and sedentary lifestyle are major risk factors for the development and progression of these chronic conditions [4]. As such, strategies to improve diet quality and increase physical activity are recommended as a core component of clinical management for people with CVD, T2D and NAFLD [5,6,7]. Many chronic disease practice guidelines have transitioned away from advice on specific macronutrient or calorie restriction to focus on the promotion of overall healthy food patterns.

The Mediterranean dietary pattern (MDP) is recommended as a cardioprotective dietary pattern with evidence of benefit across a range of chronic disease conditions [8]. The MDP is largely plant-based with high intake of wholegrain cereals, legumes, nuts, fruits and vegetables (particularly leafy greens, tomatoes, onion, garlic and herbs/spices) and liberal use of extra virgin olive oil [9,10,11]. Animal foods include fish and seafood, moderate amounts of poultry, eggs and dairy foods (particularly yoghurt and cheese) and limited red meat. In the MDP processed foods are limited, home cooking, social eating and physical activity are encouraged, and wine is consumed with meals.

A recent umbrella review of observational studies and randomised controlled trials found that greater adherence to the MDP significantly reduced the risk of overall mortality, CVDs, coronary heart disease, myocardial infarction and T2D [12]. Intervention with the MDP can improve cardiometabolic risk factors including dyslipidaemia, high blood pressure, insulin resistance, central obesity, hepatic steatosis and inflammation [13,14,15,16,17]. The MDP is recommended within practice guidelines and recent dietary consensus statements for management of CVD, T2D and NAFLD [5,6,18,19,20,21,22]. The MDP is also an appropriate dietary approach for solid organ transplant (SOT) recipients, as these patients are at increased cardiometabolic risk and CVD is established as a leading cause of death following SOT [23,24].

Whilst substantial evidence supports recommending the MDP in prevention and management of chronic disease, there can be a significant time lag for evidence from research trials to be translated into routine healthcare [25]. Cultural and geographical context is also an important consideration. Most of the large trials of MDP have been conducted in the Mediterranean region or Europe; there remains debate surrounding whether this traditional European eating pattern can feasibly be translated to non-Mediterranean settings [26,27,28]. In Australia, which has diverse multi-cultural influences, there have been a number of smaller trials conducted which largely support the evidence for feasibility and improvements in cardiometabolic risk factors with a MDP [15,29,30,31,32,33]. Many of these controlled trials involved dietitians trained to counsel on the MDP; however, it is unclear if the MDP is recommended as part of mainstream dietetics practice.

In Australia, dietitians are the only accredited practitioners for provision of medical nutrition therapy, and government healthcare plans for chronic disease management include dietitians as eligible allied health providers [34]. Whilst studies in Australia and the UK have explored dietitians’ practices for management of T2D [35,36], there was no consideration of the MDP. It has therefore not previously been explored what dietitians’ practices are in relation to recommending the MDP to patients with chronic disease.

This research project aims to understand to what extent the MDP is being routinely recommended by dietitians in Australia for patients with CVD, T2D, NAFLD and SOT recipients. It further aims to understand the barriers and enablers to incorporating the MDP in routine practice with these chronic disease patient groups.

## 2. Materials and Methods

The reporting of this study conforms to the STROBE statement for cross-sectional studies [37]. This study involved an anonymous national survey of Australian dietitians administered using the Qualtrics software (Qualtrics XM, Provo, UT, USA). The project was approved by Bond University Human Research Ethics Committee (approval HM01325). Participants were informed that consent was implied when they commenced the survey.

### 2.1. Participants

Eligible participants were dietitians who had practiced with at least one of the relevant chronic disease patient groups (CVD, T2D, NAFLD and/or SOT recipients) in an Australian healthcare setting within the past 12 months. Dietetics students who had not completed their degree were excluded.

### 2.2. Survey Development

Demographic survey questions included age (years), gender and country of birth of participant and parents. Participants were categorised as Mediterranean background if they or either of their parents were born in a country bordering the Mediterranean Sea. Dietitian data collected included main work location and healthcare setting, years since graduation, years of practice in chronic disease management and with which relevant patient groups they had practiced within the past 12 months. The survey questions related to practice of MDP were informed by the Theoretical Domains Framework. This is a theoretical framework designed to identify influences on health professional behaviour related to implementation of evidence-based recommendations [38]. Responses were on a Likert scale (five options) or multiple-choice. Participants were not provided with a definition of the MDP prior to commencing the survey. Participants self-rated their knowledge of MDP principles, confidence to describe and counsel on the MDP and whether recommending this dietary pattern is part of their role. Questions investigated whether participants agreed or disagreed there is enough evidence to support recommending the MDP to relevant chronic disease patient groups and how often participants counsel on the MDP. Participants were then asked how often they recommend individual MDP principles [39,40] related to key food recommendations, social eating and lifestyle factors which here provided a definition of the MDP and informed responses to subsequent questions. Participants also reported any education, training or self-study they had undertaken in relation to the MDP. Finally, participants were asked to rate a list of potential influences to practice based on whether they were a barrier, enabler, both or neither to recommending the MDP in their relevant practice. Throughout the survey participants were prompted to provide additional open-ended comments (optional). A draft of the online survey was piloted by a small group of dietetic students and two relevant practising dietitians and feedback was incorporated to improve readability and clarity. The final survey included a maximum of 87 question items, with some conditional release dependent on previous answers (survey questions are provided in the Appendix A).

The survey was open online from 3 to 30 November 2019. The survey invitation, including a weblink to the survey, was emailed to Accredited Practising Dietitians nationally in the Dietitians Australia (DA) weekly e-newsletter (approximately 5100 recipients) and to four of the DA Interest and Discussion Groups including, Diabetes (782 members); Dietitians in the Private Sector (769 members); Gastroenterology (806 members); and Cardiology (187 members). The survey invitation was recirculated via the DA weekly newsletter and via another four DA Interest and Discussion Groups including, Health, Behaviour & Weight Management (609 members) and Renal (337 members); and again to Cardiology and Diabetes. The survey invitation was also circulated in the Dietitian Connection weekly e-newsletter to current members (approximately 7500 Australians). Professional networks of the research team were also emailed, initiating a snowball sampling approach. There was substantial overlap between recipients of distribution methods, with the majority of Dietitian Connection Australian members also being DA members, and all of the Interest and Discussion group members, and the professional networks were DA members. A response rate was impossible to calculate as it is not recorded what number of dietitians work with our select chronic disease patient groups, and whether relevant practice was recent to meet eligibility. Once a participant had commenced the survey they had two weeks to complete the questions; after that period, any partially completed surveys were saved and stored within the survey data.

### 2.3. Data Analysis

The survey responses were exported from the Qualtrics online software to Microsoft Excel (2016, Microsoft Corp., Redmond, WA, USA), and analysed using SPSS Statistics (version 26.0, IBM, Armonk, New York, USA). Survey responses in which only the initial characteristic questions were completed were removed. The participant demographic and role characteristics as well as multiple-choice responses were analysed descriptively. Categorical variables are presented as n (%) and continuous variables as mean ± SD if normally distributed and median and interquartile range (IQR) if not normally distributed. Participant responses to how often they counsel on MDP for each of CVD, T2D and NAFLD were collapsed into two categories (never/rarely/sometimes or most of the time/always) in order to distinguish between participants who recommended the MDP routinely or not, and responses regarding their knowledge, confidence to counsel and role into three categories (strongly disagree/disagree/neither, somewhat agree, or strongly agree). Whether those categorical responses differed between the participant characteristics of: years of relevant practice experience (0–5 or 6+), work location (metropolitan or rural/regional), public or private setting, patient setting (acute/subacute or outpatient/community), and how often they personally follow a MDP (never/rarely/sometimes or most of the time/always); as well as between each other, were analysed using Chi-Square tests. Prior MDP training, education or self-study (yes or no) was also compared to participant characteristics. Statistical significance was set at *p* < 0.05. Frequency of counselling SOT recipients was not included in statistical comparisons due to smaller participants numbers.

## 3. Results

Two hundred and eleven people started the online survey, of which seven (3%) did not meet eligibility criteria as per the first screening question. A further 22 (10%) only completed initial characteristic questions and hence were removed. Therefore, 182 participants were included, of which 171 completed all relevant survey questions (84% of total commencing eligible participants). Majority of participants were female (97%), born in Australia (81%) and median age was 34 years (Table 1), which are representative characteristics of dietitians in Australia [41]. Most participants had >5 years of experience working with relevant patients and worked in a metropolitan location. Participants most often reported working in a public hospital or private community practice with relevant patients in an outpatient or community setting. Seventy-four participants (41%) selected either they or a parent were born outside of Australia, with 6% from a Mediterranean country. There was representation from across all Australian States and Territories (Appendix A) which are comparable to distribution of Dietitians nationally [42]. Data for statistical comparisons are reported in Appendix A, from which significant results are described in text. One hundred and fifteen participants provided additional optional open-ended responses throughout different stages of the survey, from which key additional findings are described.

Greater than half (62%) the participants had undertaken prior education, training or self-study on MDP (Table 1), which was most commonly reading literature or professional development outside of work. One third recalled education on MDP in their dietetics degree. ‘Other’ forms of education included Mediterranean cookbooks, family background and other dietitians. The proportion who had undertaken MDP education, training or self-study was higher in participants working in a community/outpatient compared to acute/subacute setting (70% versus 40%, *p* = 0.001) and in participants working in private versus public healthcare (75% versus 55%, *p* = 0.01). The proportion of participants who selected they ‘strongly agree’ there is enough evidence to support recommending the MDP was 62%, 46% and 39% for CVD, T2D and NAFLD patients, respectively (Figure 1). Of the smaller number of participants who practiced in SOT, at least half selected they ‘somewhat agree’ there is enough evidence for kidney (63%), heart (63%), liver (50%) and other (50%) transplant recipients. In additional comments, it was reiterated by some participants that knowledge of the MDP evidence base from peer reviewed literature or professional development was important, whereas others commented that they have limited time to keep up to date with literature and that access to education or resources with evidence summaries and direction to relevant clinical guidelines would be helpful.

Close to half the participants selected they ‘strongly agree’ they are knowledgeable of the principles of the MDP (48%), confident to describe them (48%) and confident to counsel a patient to follow them (46%), whereas 29% strongly agreed recommending MDP was part of their role (Table 2). A greater proportion of participants strongly agreed recommending MDP was part of their role if they had practiced with patients for 6+ compared to 0–5 years (37% versus 18%, *p* = 0.02) and if they worked in a private versus public setting (44% versus 20%, *p* = 0.002). Participants working in private versus public settings were also more likely to strongly agree they were confident to counsel patients on MDP (60% versus 37%, *p* = 0.01).

The proportion of participants who selected they ‘most of the time’ or ‘always’ counsel on the MDP was highest for patients with CVD (48%), followed by liver transplant (33%), NAFLD (31%), T2D (26%) and heart transplant (13%) (Figure 2). For kidney and other SOT, 50% of participants selected ‘rarely’ or ‘never’. Some participants commented that components of MDP may be integrated with other dietary counselling guidelines, including the Dietary Approaches to Stop Hypertension (DASH) diet or Australian Guide to Healthy Eating, based on their clinical judgement at the time. Responses also identified that some participants do not typically use the term ‘Mediterranean diet’ when recommending principles which align with the MDP, with comments such as *‘I don’t call it ‘Mediterranean’—I just call it a healthy eating pattern’* and *‘I rarely say “Mediterranean diet” to a client, however would encourage the principles via dietary change recommendations and small changes the client felt they were able to manage’.*

Of the dietitians who indicated they sometimes or more often counsel on MDP in at least one of the patient groups, 70% (*n* = 105/151) indicated this had been for 5 or less years. Participants were more likely to counsel regularly (i.e., most of the time/always) on the MDP if they had practiced with patients for 6+ compared to 0–5 years for CVD (58% versus 34%, *p* = 0.005), T2D (36% versus 12%, *p* = 0.001) and NAFLD (40% versus 16%, *p* = 0.01) patients. More participants also counselled regularly on the MDP if they worked in a community/outpatient compared to acute/subacute setting for CVD (55% versus 28%, *p* = 0.007) and T2D (31% versus 12%, *p* = 0.03) patients. Finally, more participants counselled regularly on MDP for patients with each of CVD, T2D and NAFLD if they strongly agreed to each of the questions related to their MDP knowledge, confidence to counsel and role.

Half the participants reported they ‘most of the time’ personally follow a MDP (Table 3). Participants were more likely to regularly counsel on MDP if they personally followed a MDP most of the time/always compared to sometimes or less for CVD (61% versus 27%, *p* < 0.001) and T2D (38% versus 9%, *p* < 0.001) patients. Participants who personally followed a MDP most of the time/always were also more likely to select they strongly agreed to each of the questions related to their MDP knowledge, confidence to counsel and role.

Majority of participants reported ‘always’ recommending to patients to limit sugary drinks, processed snacks, and processed/deli meats, and encouraging daily intake of ≥5 vegetable serves and 2–3 fruit serves (Table 3). Participants more commonly (33 to 46%) selected they ‘most of the time’ recommend fish/seafood 3+ serves weekly, wholegrain cereal 6–8 serves daily, fermented dairy foods most days, moderate consumption of wine and legumes/lentils of 2 or more serves weekly. The principles recommended least often (>50% selected ‘never’, ‘rarely’ or ‘sometimes’) were red meat of 1 or less serves weekly, intake of tomatoes daily, extra virgin olive oil 3–4 tablespoons daily, and regular use of onion and garlic, or herbs/spices in cooking. For lifestyle principles, >50% of participants ‘most of the time’ or ‘always’ encouraged home cooking, shared meals and increasing physical activity. Some participants also wrote comments that they recommend aspects but not the full MDP approach. This tended to be because their advice is dependent on the patient and with a focus on making small changes, however some also reported that their knowledge or confidence in relation to recommending cuisine, food combinations, recipes, specific proportions or social elements could be improved. Some participants also raised concerns that the MDP or specific principles such as high fat foods (liberal EVOO and nuts most days) and dairy recommendations do not align with Australian guidelines, which they prioritise.

All potential influences to practice listed in the survey were reported by the majority of participants as potential barriers and/or enablers to recommending the MDP to their relevant patients, with the exception of the participants’ own cultural background (Table 4). Most commonly reported barriers to recommending MDP to patients included number of patient visits (44%) and time allocated to consultations (38%). Some participants commented that inconsistent definitions and interpretations of the MDP as well as competing clinical priorities with complex multimorbidity can be a barrier to MDP education. In the context of post-SOT care, limited dietetics expertise in the community setting was raised.

Acceptability of the diet principles by patients was commonly reported as an enabler (33%) and ‘both a barrier and an enabler’ (32%). Most commonly reported as ‘both a barrier and an enabler’ to recommending MDP were patients’ cultural background (43%) and goals or motivation (34%). Open-ended responses gave further insights into dietitians’ perceptions of patient specific barriers to implementing the MDP. Key themes included limited cooking skills, limited time for food preparation with prioritising of convenience foods, unfamiliar foods and differing taste preferences, stage of change and lack of willingness to try, diet information overload (especially in the context of social media) and a lack of social support. Low socio-economic status and culturally and linguistically diverse patients, specifically those of Aboriginal and Torres Strait Islander or Asian cultural backgrounds, and practicing in rural and remote settings were additional barriers. Comments included, *‘Some of these recommendations are difficult to discuss due to food security, access concerns, social issues around ETOH* [alcohol], *cooking skills, access to health hardware and the cultural appropriateness of the food items being suggested’* and *‘it is often not well accepted to push the limit of red meat...Seafood [is] also a challenge in rural areas. Often unaffordable or not liked’.*

Some participants raised concerns with recommending a ‘diet’ label, for example, *‘I would never discuss these [principles] as being part of ‘a diet’, or use the words ‘the Mediterranean diet’, when talking with patients…the labelling of a ‘diet’ seems to make it less ‘real-life’/achievable/sustainable, no matter how ‘evidence-based’ the ‘diet’ is’.* On the contrary, being a ‘*diet*’ was identified as a potential enabler, ‘*Some people like to ‘follow’ a diet and the Mediterranean Diet offers that’*. Some dietitians also commented that MDP is easier to follow than other diets, palatable and can be applicable to the whole family.

Most commonly reported enablers to recommending the MDP included access to or awareness of evidence and practice guidelines (48%), professional development (45%) and patient education materials/resources (40%). Participants commented that they would like more practical based professional development and MDP focused patient education resources that are visually appealing, easily accessible and evidence based. One dietitian commented there are *‘not many useable/consumer friendly resource for teaching med diet. Have had to develop a lot of my own’* and another highlighted *‘I would like more suitable handouts to educate patients how to follow this diet with what is available and affordable in rural towns’.* Suggestions for enablers also included simple recipes, education on low cost alternatives, strategies for practical adaptation to other cultural cuisines and use of a Mediterranean diet score. It was also raised that more media attention and public health campaigns for MDP would help.

## 4. Discussion

Whilst substantial evidence supports the MDP for management of cardiometabolic diseases, it was unclear whether this dietary pattern is recommended by dietitians in non-Mediterranean settings. The current survey found an evidence-practice gap in Australian dietetics practice with fewer than 50% of participants routinely counselling patients with CVD, T2D, NAFLD and SOT on MDP. Some key principles of the MDP are frequently recommended such as increased fruit and vegetables and limiting discretionary foods, whilst other principles related to Mediterranean cuisine and limiting red meat are not routinely included in counselling. Practical application of the MDP evidence base to patients from diverse cultural backgrounds and within limited consultations were key challenges raised. Access to evidence, professional development and education resources were identified enablers.

The age and strength of evidence for use of MDP across the relevant disease groups does differ and may play a role in how frequently dietitians recommend in practice. Participants agreed there is evidence for use of MDP in CVD, followed by T2D and NAFLD populations but showed greater uncertainty in the evidence for use with SOT. This somewhat mirrored the frequency of MDP counselling to these respective patient groups. Whilst evidence alone is not adequate to initiate wide-spread changes in practice, the recognised lag time to translate emerging evidence into guidelines for clinical practice in health services may contribute to the variation in use of MDP across these disease groups. Furthermore, although observational studies of MDP in an Australian setting have existed for over two decades [43], most research trials investigating the MDP in Australian chronic disease populations have been published from 2017 onwards [44]. Some dietitians’ exposure to MDP research may have been recent and indeed those participants who counsel on MDP most often commenced this strategy within the last 5 years. Whilst there is sufficient evidence to support that following the MDP can lead to beneficial changes in cardiometabolic disease risk factors in Australians, our findings support that additional barriers may exist for clinicians to routinely recommend, or for patients to follow, the MDP. Indeed, Australian dietitians lack confidence and skills in undertaking knowledge translation [45].

In the current study, access to relevant evidence and practice guidelines was a common enabler to practicing the MDP, however, participants highlighted that keeping up to date is challenging and that direction to evidence summaries and clinical guidelines would assist. Previous studies of clinical dietitians support that limited time to source and read scientific literature, which was expected but not explicitly stated as part of their workload, is a barrier to knowledge translation [46,47]. Dietitians also perceived a lack of research matching their daily practice issues [46]. Dietitians in our study similarly commented that there may be competing clinical priorities to recommending MDP which highlights the complex nature of managing patients with chronic conditions in a ‘real-world’ setting. Controlled dietary research trials often exclude people with comorbidities or nutrient deficiencies and recruit motivated patients [48], and hence the feasibility of transferring MDP to standard practice may be challenging.

Our data found that dietitians who mainly worked in acute or subacute care were less likely to counsel CVD and T2D patients on MDP. Acute service delivery for patients with issues such as malnutrition may be prioritised over providing healthy eating education for those with chronic diseases in an inpatient setting. This aligns with literature suggesting hospital dietitians in Australia spend less than one quarter of their time in direct patient contact and only 1% providing dietary education [49]. Outpatient or community setting was more common for MDP education, however, access to dietitian services in Australia is limited [50]. Of note, SOT centres across Australia remain centralised within tertiary metropolitan hospitals in each state and specialist care within community services may be limited for transplant-related chronic disease management [51], which was raised in survey comments. Consumer engagement with liver transplant recipients demonstrated that patients have a desire to continue to engage with specialist clinicians from the transplant centres [52]. For SOT recipients, of which annual numbers have increased by 80% over the past decade [53], there is merit to consider training of dietitians more widely on long-term dietary management for reduction of cardiometabolic risk. It has also been found that NAFLD is underappreciated in primary care settings, and as methods for early detection and diagnosis advance, there is likely to be an increased demand for dietetics involvement in clinical management [54].

Frequency of counselling on MDP and agreement to being part of their role was greater in participants with 6 or more years dietetic practice experience with relevant patient groups. This could be a reflection that in non-Mediterranean countries integrating MDP with cultural and society norms, as well as deviation from national guidelines is complex [26,39] and requires additional practice training or expertise in the workforce. Furthermore, responses suggested that education on MDP was limited in dietetics degrees and largely self-directed after formal studies and more commonly accessed by dietitians working in a private setting. Integration of the MDP into university courses, including relevant clinical placements, could enhance use by dietitians earlier in practice.

The frequency of participants recommending individual principles of the MDP varied substantially. The principles most frequently recommended included promoting the intake of fruit and vegetables and limiting intake of processed snacks, processed meats and sugary drinks. These principles are aligned to healthy eating patterns in general, including the Australian Guide to Healthy Eating [55] as well as other evidence-based diets participants use to guide practice such as the DASH diet, and are likely most familiar to clinicians as well as consumers.

Principles which were less frequently recommended included substantial reduction in red meat and regular intake of plant-based protein sources, daily intake of tomatoes, regular use of onion, garlic and other herbs and spices and liberal use of extra virgin olive oil. These are core principles which distinguish a MDP from other diets, and they make important contributions to its anti-inflammatory and antioxidant properties [56,57,58,59]. These more specific recommendations may be a lower priority to cover with patients who have poor baseline dietary habits but also appear to be influenced by the dietitian’s own confidence and experience to incorporate these dietary principles. Recommendations to limit total dietary fat intake have only recently been removed by the National Heart Foundation [60] and are arguably still promoted in the Australian Guide to Healthy Eating (released 2013) [55], which may contribute to hesitation to recommend larger quantities of extra virgin olive oil and nuts, particularly for chronic disease management which traditionally prioritises weight reduction as a goal of treatment.

A range of perceived patient-centred barriers and enablers to recommending the MDP in terms of its principles but also its name were raised. Indigenous Australian and Asian cultural backgrounds, familiarity, preparation skills and convenience were among the key barriers. Perceptions around the influence of motivations, acceptability of the diet, social support and health literacy appeared to be more dependent on the individual patient situation. Whilst clinicians’ perspectives of barriers and enablers to recommending MDP have not previously been explored, studies have been done with consumers involved in intervention trials of a Mediterranean diet. Focus groups in adults in the United Kingdom highlighted the MDP to be enjoyable and pleasurable but they had difficulties purchasing food items and work, stress and time pressures undermined adherence [28]. Interviews with patients with NAFLD in Northern Europe revealed that life stressors and demand for convenience were challenges to MDP uptake, whereas facilitators included increased nutrition knowledge and skills, family support and MDP promotion in media and clinical settings [61]. Focus groups with liver transplant recipients in Australia also identified employment as a competing priority, however they described a positive experience with adapting their diet to a Mediterranean-style of eating [62]. Individuals in Northern Europe at high CVD risk perceived that barriers to adopting the MDP would be (i.e., they had not actually undertaken an intervention) living in a colder climate, impact on body weight, acceptability and cultural differences [63]. It has also been shown that increased media coverage and inclusion in books is an enabler for MDP adoption [61,64], which aligns with dietitians comments.

Over a third of participants identified as working in regional or rural areas, which have well accepted health disadvantages associated with higher prevalence of chronic disease, inequity of health care access and greater food insecurity compared with metropolitan areas [65,66]. Cost of food was more frequently raised as a barrier by participants from these areas. When the concept of diet cost is challenged in a research setting, the MDP consistently reveals to be feasible and no more expensive than usual eating habits, through both cost analysis [67,68] and from patient perspectives [61,62]. Many participants in this survey indicated cost was neither a barrier or enabler. However, there remains challenges recommending this diet to patients who may have heightened food insecurity or limited utilities of skills in food preparation, irrespective of location.

Strategies to develop practical skills and resources was identified as a key enabler for dietitians to integrate MDP into therapeutic counselling. Whilst a background understanding of the evidence and core dietary principles is important, it appears the evidence-practice gap is particularly linked to a lack of patient education resources with practical application of modifying Westernised diets to a more Mediterranean style pattern. It was suggested that access to more materials which are simple, pictorial and include low cost recipes and culturally appropriate options for diverse populations, would help. Similarly, process evaluation of a healthy eating website intervention promoting the Mediterranean diet for women in Scotland found that the most regularly visited sections were those providing practical advice, including recipes [69]. Furthermore, our study indicates conflict remains regarding acceptable terminology for labelling the MDP with some dietitians expressing the use of the word ‘diet’ was inappropriate when counselling on healthy eating.

Knowledge sharing by dietitians with research and clinical MDP expertise is important and should include a focus on strategies for practice and ideally case studies. It is important for these opportunities to be equally available to clinicians in regional and rural settings and online forums could be an important platform. There is a growing base of practical literature and resources from Australian dietitians [39,70,71,72,73] but wider dissemination across Australia is needed, ideally including sharing of materials developed for clinical trials which have been rigorously evaluated. Australia could also adapt and evaluate evidence-based tools from other non-Mediterranean countries, such as Oldways from the United States [74], and the Dietitians of Canada Mediterranean diet toolkit [75]. In the context of limited public access to dietitians for chronic disease management, strategies for ongoing support to patients to implement long-term behaviour change required to sustain a MDP are needed. Enhanced use of technology and telehealth [76] as well as training of other primary care practitioners [77] could be useful strategies.

This study is strengthened by its national reach and representation of dietitians across Australian states, work locations, healthcare services and patient settings. Furthermore, within the eligible dietitians who started the survey the completion rate of survey questions was greater than 80%. There are limitations however in knowing the true response rate, particularly among the target group as it is unknown what proportion of dietitians in Australia work with relevant patient groups. It is also unknown whether the participants are truly representative of dietitians specifically working with these relevant patient groups in Australia. There was also a potential for respondent bias in that those who participated in the survey may have a specific interest in the MDP, which may have resulted in an underrepresentation of the evidence-practice gap.

## 5. Conclusions

An evidence-practice gap exists in Australian dietetic practice with less than half of participants routinely counselling the relevant chronic disease patient groups on the MDP. Strategies to support dietitians to counsel on the MDP within limited consultations are needed. This should include available in-person and online education and consumer-friendly practical resources that are accessible to student and clinical dietitians in varying work locations and settings. Furthermore, using knowledge translation frameworks, there is a need for long term multi-site evaluation of implementation of these MDP resources in routine practice including use by dietitians and effectiveness for patient adoption of dietary principles and clinical outcomes.

## Figures and Tables

**Figure 1 nutrients-12-03395-f001:**
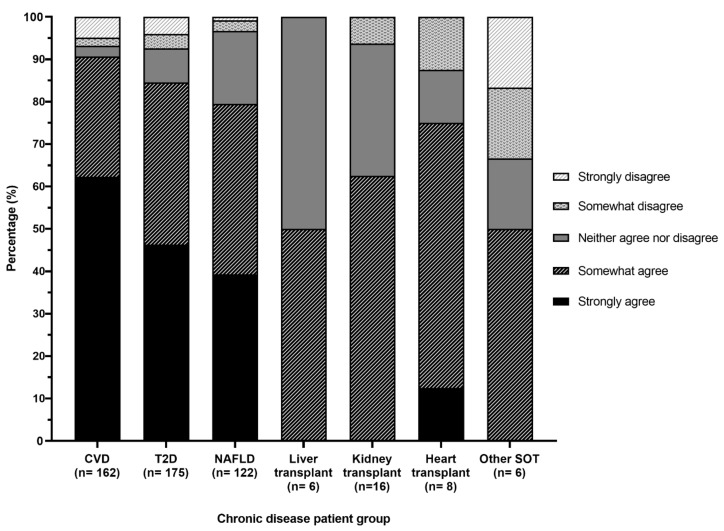
Level of agreement regarding evidence to support the Mediterranean dietary pattern across chronic disease populations. CVD, cardiovascular disease; T2D, type 2 diabetes; NAFLD, non-alcoholic fatty liver disease; SOT, solid organ transplant.

**Figure 2 nutrients-12-03395-f002:**
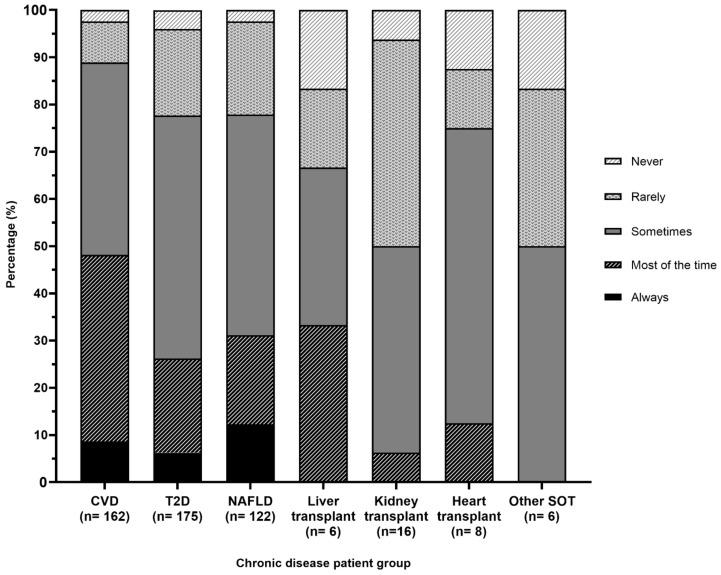
Frequency of dietitians counselling on the Mediterranean dietary pattern across chronic disease populations. CVD, cardiovascular disease; T2D, type 2 diabetes; NAFLD, non-alcoholic fatty liver disease; SOT, solid organ transplant.

**Table 1 nutrients-12-03395-t001:** Characteristics of dietitian participants (*n* = 182).

Characteristics	N (%)
Gender (female)	177 (97)
Age (years), median (IQR)	34 (29–44)
Born in Australia	146 (81)
Mediterranean background ^a^	11 (6)
Time since graduating as a dietitian (years), median (IQR) (*n* = 180)	9 (4–18)
<1	10 (6)
1–5	50 (28)
6–10	44 (24)
11–15	23 (13)
16–20	20 (11)
20+	33 (18)
Duration practicing with relevant chronic conditions (years)	
<1	9 (5)
1–5	64 (35)
6–10	47 (26)
11–15	24 (13)
16–20	14 (8)
20+	24 (13)
Main work location	
Metropolitan	120 (66)
Regional	39 (21)
Rural	23 (13)
Main place of work	
Public Hospital	68 (37)
Private Hospital	6 (3)
Public Community Health Service	39 (21)
Private Community Practice	62 (34)
Aged Care	2 (1)
Corporate Health	2 (1)
Non-Government Organisation	3 (2)
Main patient setting	
Acute	36 (20)
Sub-acute	10 (6)
Outpatient	70 (39)
Community	66 (36)
Prior education, training or self-study on MDP (*n* = 178)	110 (62)
Dietetics degree ^b^	53 (30)
Professional development provided at work	21 (12)
Professional development accessed outside of work (not a conference)	71 (40)
Reading scientific literature	83 (47)
Australian conference	37 (21)
International conference	2 (1)
Research activities	20 (11)
Other	8 (4)

MDP, Mediterranean dietary pattern. ^a^ Participant or parent born in a country which borders the Mediterranean Sea. ^b^ Participants selected all applicable options for type of prior education, training or self-study. Note: responses may not equal 100% due to rounding.

**Table 2 nutrients-12-03395-t002:** Participants’ perceived knowledge, confidence and role in relation to counselling on the MDP (*n* = 182).

Question	Strongly Disagree	Somewhat Disagree	Neither Agree Nor Disagree	Somewhat Agree	Strongly Agree
			N (%)		
I am knowledgeable of the principles of the MDP	3 (2)	1 (1)	5 (3)	86 (47)	87 (48)
I am confident to describe to a colleague the key principles of the MDP	2 (1)	5 (3)	5 (3)	83 (46)	87 (48)
I am confident to counsel a patient to follow the principles of the MDP	2 (1)	7 (4)	11 (6)	78 (43)	83 (46)
Recommending the MDP to patients with chronic disease is part of my role	4 (2)	11 (6)	32 (18)	82 (45)	53 (29)

MDP, Mediterranean dietary pattern. Note: responses may not equal 100% due to rounding.

**Table 3 nutrients-12-03395-t003:** Participants’ responses to how often they recommend (in their advice, education or resources) individual principles of the MDP to relevant chronic disease patients (*n* = 178).

MDP Principle ^a^	Never	Rarely	Sometimes	Most of the Time	Always
	N (%)				
Daily intake of fruit (2–3 serves)	1 (1)	3 (2)	19 (11)	53 (30)	102 (58)
Daily intake of vegetables (5 or more serves)	1 (1)	2 (1)	10 (6)	36 (20)	129 (73)
Daily intake of tomatoes	43 (24)	52 (29)	55 (31)	22 (12)	6 (3)
Daily intake of leafy green salads/vegetables	9 (5)	12 (7)	37 (21)	56 (32)	64 (36)
Regular use of onion and garlic in cooking	30 (17)	45 (25)	63 (35)	32 (18)	8 (5)
Regular use of herbs and spices in cooking	10 (6)	21 (12)	63 (35)	57 (32)	27 (15)
Daily intake of wholegrain cereals (6–8 serves)	3 (2)	10 (6)	37 (21)	72 (40)	56 (32)
Use of extra virgin olive oil as the main dietary fat	1 (0.6)	6 (3)	37 (21)	52 (29)	82 (46)
Daily intake of extra virgin olive oil 3–4 tablespoons per day	33 (19)	47 (26)	56 (32)	28 (16)	14 (8)
Intake of fermented dairy foods (yoghurt and cheese) on most days	12 (7)	21 (12)	51 (29)	60 (34)	34 (19)
Limit intake of red meat to no more than 1 serve per week	18 (10)	49 (28)	56 (32)	46 (26)	9 (5)
Limit intake of processed/deli meats and small goods	1 (1)	4 (2)	10 (6)	64 (36)	99 (56)
Regular intake of legumes/lentils (2 or more serves per week)	0	17 (10)	50 (28)	59 (33)	52 (29)
Regular intake of fish/seafood (3 or more serves per week) ^b^ (*n* = 95)	0	2 (2)	16 (17)	44 (46)	33 (35)
Intake of nuts on most days	1 (1)	14 (8)	51 (29)	64 (27)	48 (27)
If choosing to drink alcohol, moderate consumption of wine (1–2 glasses per day) with meals	7 (4)	25 (14)	38 (21)	60 (34)	48 (27)
Limit intake of sugary drinks	0	1 (1)	7 (4)	31 (17)	139 (78)
Limit intake of commercial or processed sweets/goods/savoury snacks	2 (1)	0	6 (3)	46 (26)	124 (70)
Encourage eating with others or shared meals	5 (3)	22 (12)	59 (33)	47 (26)	45 (25)
Encourage home cooking	1 (1)	1 (1)	19 (11)	79 (44)	78 (44)
Increasing regular physical activity (*n* = 170)	1 (1)	6 (4)	46 (27)	67 (39)	50 (29
Would you estimate that you personally follow a MDP?	2 (1)	4 (2)	66 (37)	91 (51)	15 (8)
How often do other dietitians that you work with or know recommend the MDP to patients with chronic disease? *N = 81 (45%) responded ‘I don’t know’*	2 (1)	8 (4)	62 (34)	24 (13)	5 (3)

MDP, Mediterranean dietary pattern. ^a^ Serve sizes were advised to be reflective of the Australian Dietary Guidelines. ^b^ Question added 10 days after survey open date due to an error in original survey. Note: responses may not equal 100% due to rounding.

**Table 4 nutrients-12-03395-t004:** Barriers and enablers to whether participants recommend the MDP to their relevant patients (*n* = 171).

Question	More Often a Barrier	More Often an Enabler	Both a Barrier and an Enabler	Neither a Barrier or an Enabler	Not Relevant to My Practice
	N (%)				
Access to or awareness of relevant evidence/practice guidelines	22 (13)	82 (48)	24 (14)	42 (25)	1 (1)
Access to or awareness of professional development, education or training for dietitians	36 (21)	77 (45)	23 (14)	33 (19)	2 (1)
Access to or awareness of relevant patient education materials or resources	52 (30)	69 (40)	20 (12)	28 (16)	2 (1)
Acceptability of the diet principles by patients	38 (22)	56 (33)	54 (32)	21 (12)	2 (1)
Goals or motivation of patients in relation to diet	44 (26)	49 (29)	58 (34)	19 (11)	1 (1)
Your own cultural background	7 (4)	53 (31)	15 (9)	82 (48)	14 (8)
Cultural background of your patient	47 (28)	21 (12)	74 (43)	27 (16)	2 (1)
Support from/goals of the multi-disciplinary team or other clinicians treating your patient	17 (10)	45 (26)	38 (22)	56 (33)	15 (9)
Reason for dietitian referral	19 (11)	49 (29)	40 (23)	56 (33)	7 (4)
Time allocated to patient consultations	65 (38)	25 (15)	15 (9)	60 (35)	6 (4)
Number of patient visits	75 (44)	24 (14)	15 (9)	50 (28)	7 (4)
Cost of the diet	49 (29)	19 (11)	40 (23)	59 (35)	4 (2)

MDP, Mediterranean dietary pattern.

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
