# Peer review of "Investigating Whether the Mediterranean Dietary Pattern Is Integrated in Routine Dietetic Practice for Management of Chronic Conditions: A National Survey of Dietitians"

_nutrients, 2020, doi:10.3390/nu12113395_

Round 1

Reviewer 1 Report

This is an important paper and directly related to practice, which would be of great interest to the journal’s audiences. The paper is well written, with appropriate methods. The findings are appropriately discussed, limitations are acknowledged and the conclusions reached are valid.

Introduction:

  1. Line 52. Please replace ‘feta cheese’ to just ‘cheese’. To my knowledge, none of the papers describing the MD refer to feta cheese in particular, and also feta cheese is Greek and therefore this definition would not cover other Med regions that made the MD known, such as Southern Italy.
  2. Line 73. Please replace ‘is being routinely practiced’ with ‘is being routinely recommended (or prescribed)’.
  3. I think the Introduction would benefit from a short paragraph describing the norms of the dietetics practice in Australia. Things to consider including is what the Australian Dietetic Association guides dietitians to do and whether patients are referred to dietitians as part of their routine care or if it’s common for them to obtain dietetic advice elsewhere. It will also be useful to discuss other studies conducted among dietitians – the group by McArdle has done a lot of work on this in the UK (also a non-Med country), albeit for type 2 diabetes.

Methods

  1. Was the definition of the MD provided to dietitians as part of the survey? I notice in the abstract you report on the recommendation of onions/garlic, tomatoes and olive oil – which is the sofrito question of the questionnaire used in the PREDIMED index. It’s important to acknowledge whether dietitians were indeed aware of what the MD is, to ensure valid responses, and which instrument questions 44-63 of your questionnaire are based on. For example, I am not aware of any definition of the MD that suggests a daily intake of tomatoes.
  2. Thank you for providing the questionnaire in the Supplement, for replication purposes.
  3. Lines 124-125. I am unsure what ‘Partially completed surveys were saved and stored within the survey data after two weeks’ means. Can you please clarify.
  4. Lines 133-134. What led to the decision for responses to how often dietitians counsel on MD to be collapsed into two categories (never/rarely/sometimes or most of the time/always)? Surely ‘sometimes’ is different to ‘never/rarely’ and I would like the authors to justify this decision please.

Results:

  1. The Results are very well written and present a comprehensive and detailed report of the findings. I found some findings really interesting, e.g. Table 4, 10% of participants reported that the cost of the diet would be an enabler to them recommending the diet to their patients. Also, I wonder why 14% thought that the number of visits would be an enabler. It would be really interesting if the authors added a table in the supplement with more of the participants’ quotes in the open-ended questions. I think this would enrich the manuscript further and give deeper meaning to the quantitative data.

Discussion:

  1. L295. Please also include the main findings of enablers, as this was one of your research questions.
  2. L304-306. I find this statement very interesting and I think I disagree. I am not sure whether it might be due to the large Greek (and potentially other Med?) community in Australia (at least in urban areas?), but there have been many papers on MD adherence in Australia, led by the Kouros-Blazos team (I remember one of their papers from 1999 that I cited in my own MSc dissertation!). I agree however, that RCTs have been conducted only more recently, but the authors should acknowledge that surely the MD is not a new concept?
  3. Also, there is sufficient evidence that the MD is associated with less disease risk, and can lead to beneficial changes to chronic disease risk factors (e.g. metabolic risk factors). It is unlikely that the Australian population will not have the same benefits if they follow the MD, so I would not see this as a reason for having a gap between evidence and practice, and I would like to ask the authors to acknowledge this. However, there is enough evidence from the UK (papers from Papadaki, Woodside) to support your concerns that potential acceptability of the diet by the public/patients might indeed be problematic.
  4. Line 346. Please replace ‘participants’ with ‘practitioners’, as there is no indication that you questioned participants about other dietary patterns and/or the DASH diet.

Reviewer 2 Report

This was a paper that I truly enjoyed reading. It has a logical sequence, it flows well and the authors seemed to have read their background literature. 

I have some comments for the authors, which I would like them to consider:

1) Introduction: Indeed one of the main questions is whether you can implement the MDP in a non Mediterranean environment. I am surprised however, that in your introduction you don't mention even one of the successful U.K. studies conducted by Klonizakis and colleagues. These were not big trials, but surely there was a good indication of the potential (for example see Liu et al., Nutrition. 2018 Nov;55-56:185-191) and you refer to these.

2) Participants: You will need to provide some more information on the participants and the type of healthcare settings that you included. I did see later on (Survey Development) that you primarily had participants from the private sector (which is understandable), but we do need to see if you covered all regions and settings of Australia, as this would effect the overall outcomes. For example you would expect them to suggest MDP easier in Melbourne where there are about 500,000 Greeks, but is it the same in Perth? If not this needs to be discussed as a limitation at the end (apparently you don't seem to have found any, which is impossible to be the case). 
Another question that springs to mind (and is also related to the limitations' aspect) is whether your findings are truly representing the population that you intend to study. I don't object the fact that you worked with "the average" but I would have liked to see something more convincing e.g., a sample size calculation or an estimation, as well as effect sizes (if it was possible to make). Again, the fact that you got 182 out of the 5100 makes it about 4%, which compared to the lack of statistical cover makes it a limitation, which should be added at the end of the manuscript. 

3) Discussion. I have to be a bit sceptical about your suggestion to share for Australians to share tools with other countries. In U.K. for example there is not a similar programme being implemented and apart from some online presentations of the diets (and apart from the work by Klonizakis and colleagues discussed earlier), there is not a programme being implemented elsewhere in a formalised basis. This also contradicts your original suggestion that not much is known about whether it is possible to use it in non Mediterranean settings, so I would suggest playing this down a bit. 

4) Conclusions. This paper will be really strengthened if you suggest a way forward. I would like to see some actions and certainly a suggestion of a long-term, multi-centre study to explore the efficiency and efficacy of a MDP programme.

Reviewer 3 Report

This is an interesting study with good scientific quality and good presentation. The paper needs no improvement

Round 2

Reviewer 1 Report

Thank you for responding to my suggestions. I am very happy to see this paper published; congratulations on this very important work.